# New Poplar-Derived Biocomposites via Single-Step Thermoforming Assisted by Phosphoric Acid Pretreatment

**DOI:** 10.3390/polym14173636

**Published:** 2022-09-02

**Authors:** Deming Chen, Chunyan Xu, Haoran Ye, Yang Shi, Yequan Sheng, Shengbo Ge, Minglong Zhang, Hui Wang

**Affiliations:** 1College of Materials Science and Engineering, Central South University of Forestry and Technology, Changsha 410004, China; 2Jiangsu Co-Innovation Center of Efficient Processing and Utilization of Forest Resources, International Innovation Center for Forest Chemicals and Materials, College of Materials Science and Engineering, Nanjing Forestry University, Nanjing 210037, China; 3Aerospace Kaitian Environmental Technology Co., Ltd., Changsha 410000, China; 4Hunan Hongsen Novel Material Technology Co., Ltd., Yiyang 413407, China

**Keywords:** poplar powder, pretreatments, hot-moulding, biocomposites, high performance

## Abstract

One-step thermoforming represents an effective approach to preparing glue-free biocomposites. This study aimed to produce glue-free biocomposites with high-temperature resistance and mechanical properties using phosphoric acid pretreatments combined with thermoforming. Due to the hot-moulding process, the cell wall was destroyed, which allowed the fibres to adhere closely together. Most hemicelluloses were hydrolysed through pretreatment with phosphoric acid, and the contact area between the cellulose and lignin was significantly increased. The biocomposites prepared by ball milling demonstrated remarkable flexural strength (49.03 MPa) and tensile strength (148.23 MPa). Moreover, they had excellent thermal stability, with the maximum temperature for pyrolysis rate at 374 °C, which was much higher than that of poplar (337 °C). In addition, the material released no formaldehyde during the preparation process, which is in line with the concept of green production.

## 1. Introduction

Biocomposites are materials with excellent performance made of renewable resources through biological, chemical, and physical methods or composited with other materials. Petroleum-based composites were commonly used in the past due to the low price of petroleum [1]. Comparatively, biocomposites are not used as widely as petroleum-based composites because of the low economic potential of the materials [2]. However, due to the ever-increasing demands of people today, composite materials are in increasing demand [3]. Synthetic additives bond traditional biocomposite materials after the biomass materials are moulded. Among the most common types of biocomposite are plywood, particleboard, and fibreboard, as well as various deep-processed materials that allow the wood to be applied in different ways and result in effective resource conservation. Biocomposites have also begun to attract the attention of scientists. They are used extensively in architecture, furniture, and decoration [4,5,6]. Unfortunately, the synthetic adhesives used for preparing traditional biocomposite materials tend to release formaldehyde, which is harmful to the environment. Currently, there is no perfect solution for eliminating the health risks associated with formaldehyde release, which means that the use of traditional biocomposites in furniture manufacturing will be severely restricted [7]. Furthermore, the International Agency for Research on Cancer has classified formaldehyde as a Category A1 human carcinogen [8,9]. Therefore, it is necessary to explore a practical yet economically feasible process for preparing formaldehyde-free biocomposites [10].

The development of formaldehyde-free adhesives is the key to creating a formaldehyde-free biocomposite. Polymethylene Diphenyl Isocyanate (PMDI) is used as an aldehyde-free adhesive to produce formaldehyde-free fibreboard. The production of PMDI, however, is dependent on petrochemical resources. If PMDI is used extensively, it will result in the rapid consumption of petroleum resources, which is not in line with the notion of sustainable development. For this reason, some scientists are devoting attention to developing bio-based adhesives. For example, a good-quality bio-adhesive was prepared by combining NaOH-activated lignin with natural rubber latex [11]. However, due to technical limitations, existing formaldehyde-free board production methods are too costly and complex to be commercially viable [12,13,14]. This paper proposes a green biocomposite (self-adhesive materials) alternative to traditional biocomposite. This proposed material will not require adhesives for binding, thereby reducing production costs. Biocomposites produced by self-adhesive technology have superior dimensional stability without the use of adhesives. Many researchers have explored different preparation methods for non-adhesive wood products, including hot pressing. For example, bamboo biomass was hot-pressed into a biocomposite without the use of adhesive for construction and furniture applications [15]. Zhu et al. examined the relationship between the toughness and strength of cellulose nanopaper and the size of the constituent fibres. According to their findings, the cellulose nanopaper became tougher and stronger as fibre size decreased [16].

The composition of wood primarily consists of lignin, hemicelluloses, and cellulose. In particular, lignin can act as a “natural binder” due to its excellent adhesive properties. Researchers have found that lignin can be used as a biocomposites binding agent as an alternative to traditional synthetic binders by enhancing the strength and stiffness of the resulted biocomposite structures [17,18,19]. However, the hemicelluloses were dispersed between lignin and cellulose, reducing the contact area between these components. There are many free hydroxyl groups in hemicelluloses, which are connected to one another by β-1,4-glycosidic bonds. Hence, they are characterised by undesirable high hydrophilicity [20]. Studies have shown that the densification of wood by partial removal of hemicelluloses and lignin can enhance the properties of biocomposites.

Additionally, the removal of hemicelluloses promotes a tighter arrangement of fibres and increases the density of the material. It has also been reported that the glycosidic bonds of hemicelluloses could be easily broken in an acidic medium to degrade them [21]. However, the information on the chemical and physical properties of biocomposites after the removal of hemicelluloses is limited. Hence, this study revealed the properties of biocomposites produced by hot pressing with 40–60 mesh poplar through a series of pretreatments with diluted phosphoric acid. The innovation of this experiment was the use of phosphoric acid pretreatment to remove hemicelluloses, which improved the mechanical properties, thermal stability, and hydrophobicity of the biocomposite.

This study uses phosphoric acid pretreatment to explore how aldehyde-free biocomposite materials can be made practical through phosphoric acid pretreatment. Firstly, biocomposites were prepared by pretreatment and hot-moulding (Figure 1). Then, the physical properties, such as water resistance, mechanical properties, and heat resistance of biocomposites were tested. In addition, the chemical composition and surface of the material were characterised. Finally, the mechanical properties of traditional fibreboard were compared with the resulted biocomposite to highlight its practicality as a potential good-quality material for construction and furniture manufacturing.

## 2. Materials and Methods

### 2.1. Source of Feedstock and Its Utilisation to Produce Biocomposites 

The poplar powder (40–60 mesh) used in this study was supplied by the Lianyungang timber wholesale market in Northern Jiangsu. Then, the powder was pre-treated under different conditions, such as phosphoric acid soaking, microwave treatment after soaking, and ball milling with a phosphoric acid solution (Appendix A). First, the poplar wood flour was soaked in 5% phosphoric acid for 3 h, and the pre-treated poplar wood powder was rinsed with plenty of water until the residual phosphoric acid was completely removed. Then, powder samples were dried in an oven at about 100 °C for one day. About 8 g of powder were taken from each sample and followed by subjecting to pressurised hot-moulding (30 MPa) for 60 min at 186 °C with a ZG-50TSD thermocompressor manufactured by Zhenggong Electromechanical Equipment Technology Co., Ltd., located in Dongguan, China, to synthesise a 25 cm^2^ biocomposites with a thickness of 3.5 mm. Each sample produced three parallel specimens. Furthermore, only one parameter of hot-moulding (30 MPa, 186 °C) was selected as the control variable.

### 2.2. Analysis of Essential Mechanical and Physical Characteristics of Biocomposite

The water absorption, flexural strength, tensile strength, and density of biocomposites were tested using the GB/T 17657-2013 standard. The universal testing machine (AGS-X, Shimadzu, Kyoto, Japan) was used to test the tensile and flexural strengths of biocomposites. The tensile and flexural strengths of biocomposites were tested by applying tensile stress and bending force, respectively. The biocomposites were immersed in water, and the changes in weight and thickness were measured every other day until the water absorption was saturated. Each sample was tested three times and the average value is reported. Moreover, the water absorption of the biocomposites was evaluated via complete panels while the biocomposites were sliced 8 mm wide to quantify the mechanical characteristics. 

DSA100S contact angle tester manufactured by KRUSS in Germany was adopted to analyse the contact angle of biocomposites at various periods, including 0, 2, 5 and 10 s. About 10 mg powder of biocomposites was taken to test the thermal stability by heating the sample steadily at 30 °C/min from 20 °C to 800 °C in a TGA55 thermogravimetric analyser manufactured by TA Instruments company located in Texas, United States. The thermal conductivity of the samples was tested with a DRPL-2B thermal constant analyser manufactured by Xiangtan Instrument Co., Ltd. in China. These samples (20 mm × 20 mm × 3 mm) were fixed on the hot and cold poles of the thermal conductivity tester, the sensor determined the temperature, and the instrument analysed the thermal conductivity of the sample. A laser generator (VCL-808nmM1-7W, Blueprint Co., Ltd., Beijing, China) was used to irradiate the boards and an infrared camera (FLIR E5, FLIR Systems, Inc., Wilsonville, OR, USA) was used to capture the temperature change on the surface of the boards. The laser power was 0.5 W, and the spot diameter was 10 mm. The laser irradiation position on boards remained for 5 min, and every board was photographed every 1 min. Each sample (width 5 mm) was burnt with an alcohol lamp for 10 s, and the burning progress was recorded by video. The surface and sectional morphologies of the biocomposites were analysed with the Quanta 200 scanning electron microscope (SEM) manufactured by the FEI company in China. The surface colour of the bio-composites was analysed with an RM200QC chromatic meter manufactured by Shanghai Jintian Instrument Co., Ltd. in Shanghai, China. 

### 2.3. Chemical Structure Characterisation

The biocomposites were examined by an X-ray diffractor manufactured (XRD) by Beijing Purvey General Instruments Co., Ltd. in Beijing, China, to measure the crystallinity. A Nicolet IS50 FT-IR spectrophotometer manufactured by Thermo Fisher Technologies, Waltham, Massachusetts, in the USA, was utilised to examine chemical functional groups under a wavelength range of 400–4000 cm^−1^. The lignocellulosic content of biocomposites was interpreted via Waters E2695 high-performance liquid chromatography. The acid hydrolyzate of biocomposites was analysed by high-performance liquid chromatography (HPLC) to test the content of each component of lignocellulose.

## 3. Result and Discussion

### 3.1. Physical Properties

As shown in Figure 2a, the density of biocomposites (1.26–1.36 g/cm^3^) was higher than that of poplar (0.40 g/cm^3^), which the reduction of pores may cause. This corroborates the SEM micrographs, where large pores were found in the cross-section of poplar, while the cross-section of biocomposites was dense and smooth (Figure 2b,c). This proves that the cell wall collapses in the biocomposite, the pores are flattened, the distance between cells decreases, and the tight junctions between fibres form chemical bonds after hot-pressing [22,23]. Biocomposites containing less porosity have a higher mass per volume of fibres and a higher density [24], which may improve the mechanical properties of biocomposites. As shown in Figure 2e,f, the tensile strength and modulus of biocomposites were higher than those of poplar, which may be caused by the increase of density and the enhancement of bonding between fibres.

Furthermore, biocomposites exhibited significantly lower flexural strength and modulus than poplar, which might be attributed to the decrease in the length of fibres. The highest tensile strength of bio-composite prepared by wood flour with ball milling (WF(PA/B)) was 148 MPa, about three times higher than that of poplar. It is worth noting that the flexural strength of biocomposites prepared by wood flour with phosphoric acid (WF(PA)) was lower than that of biocomposite prepared by wood flour (WF) (Figure 2h,i). Therefore, the phosphoric acid pretreatment method did not always improve the properties of biocomposites, and the increased strength was often accompanied by increased brittleness. In addition, compared with medium-density fibreboard (MDF) and high-density fibreboard (HDF) (Figure 2d,g), the tensile strength and flexural strength of WF(PA/B) greatly improved. Therefore, the biocomposite can replace traditional wood-based panels (MDF, HDF) and be applied to furniture and decoration materials [25]. By comparing the specific strength between the prepared biocomposite and other commonly used materials (Figure 2j), it was observed that the resulted biocomposite had higher specific strength than many other materials such as metal, alloy, bricks, and cement. This indicates that the biocomposites had the conditions for application in structural building materials of specific strength [26,27].

### 3.2. Hydrophobicity of Raw Material and Biocomposites 

The water resistance of poplar and biocomposites was tested to characterise their hydrophobic properties. Based on the contact angle, biocomposites showed a significantly higher degree of hydrophobicity than poplar, while WF(PA/B) showed the greatest coefficient of hydrophobicity (Figure 3a,b) [28]. As shown in Figure 3c, water absorption by biocomposites was lower than poplar, with WF(PA/B) having the lowest rate (48.87%) after soaking for 72 h. The hot-pressing process of the fibres caused the pores of the biocomposites to become more tightly packed, thereby preventing water from penetrating the pores [29]. At the same time, lignin covered the biocomposite surface during hot-pressing, which hindered water entry to a certain extent [30,31]. In addition, the water absorption of WF(PA/B) was lower than that of WF (Figure 3c), indicating phosphoric acid treatment can improve dimensional stability and hydrophobicity of biocomposite materials, probably by reducing hemicelluloses of wood powder. It is worth noting that the shape of WF(PA) and biocomposite prepared by wood flour with microwave process (WF(PA/M)) changes immediately after immersion in water, as opposed to WF and WF(PA/B), which remained unchanged after immersion for three days (Figure 3d). In conclusion, WF(PA/B) has the potential to be used in humid environments.

### 3.3. Stability of Biocomposites against High Temperature

The thermal stability and thermal conductivity of the biocomposites were tested. As shown in Figure 4a, the poplar weight was first reduced, while the WF(PA/B) weight was reduced at higher temperatures in the main thermal decomposition stage (240–380 °C). This may be due to the WF(PA/B) having the lowest hemicellulose content, with the hemicelluloses having the lowest degradation stability among cellulose hemicelluloses and lignin [32,33]. The DTG curve further corroborates this finding, where the poplar containing more hemicelluloses tended to degrade first at lower temperatures in 240–380 °C. The mass-loss rate of poplar was the highest, whereas the WF(PA/B) was the lowest in the temperature range of hemicelluloses (200–280 °C). The mass-loss rate of WF(PA/B) was the higheest at 350–400 °C, which may be caused by the higher proportion of cellulose and lignin, requiring higher decomposition temperature to degrade these components.

The maximum decomposition rate temperature of WF (364 °C) was higher than that of poplar (337 °C), which may be attributed to the high-temperature carbonisation film formed by lignin on the surface of biocomposites by hot pressing, which subsequently improved the thermal stability of biocomposites [34]. The residual amounts of WF(PA) and WF(PA/M) were the lowest at 600 °C, which may be caused by the relatively low lignin content of both (Figure 4b). Biomass with higher lignin content has a higher residual amount after thermal decomposition [35,36]. The biocomposites prepared by hot pressing showed remarkable thermal stability in the combustion experiment (Figure 4e). Compared with other pretreatment methods, the heat resistance and thermal stability of WF(PA/B) were the best. There may have been a protective layer formed when the lignin was transferred to the surface of the board during hot pressing. During high-temperature combustion, part of the hemicelluloses was decomposed, and lignin formed a dense carbonisation film on the surface of bio-composites, which hindered the decomposition of fibres from improving the thermal stability and thermal degradation resistance of biocomposites [37,38].

The comparative analysis of thermal images showed that WF(PA/B) and WF were heated to higher temperatures with a more uniform temperature distribution in the same time [39] (Figure 4d). This shows that they were isotropic in heat transfer, while natural poplar was anisotropic [40]. The WF edge affected the whole thermal conductivity due to the thermal deviation during hot-pressing, and this error needed to be excluded from the analysis. In addition, the biocomposites did not exhibit any noticeable change in their thermal conductivity after the phosphoric acid treatment (Figure 4c). WF(PA/B) is more suitable for high-temperature working environments than other biocomposites.

### 3.4. Micromorphology Investigation

SEM images (Figure 5a,c,e) show that the surface of biocomposites was dense and smooth, while the surface of poplar is loose and porous (the cracks could be caused by hot pressing). The smooth and dense surface was covered with a lignin layer, which effectively reduced water infiltration and significantly improved the hydrophobicity of the surface and the dimensional stability of the biocomposite. The surface micromorphology of WF(PA) and WF(PA/M) (Appendix A) shows that, compared with WF, the fibres treated with phosphoric acid were challenging to combine, and the surface of the board was rougher, which decreased the surface hydrophobicity [41,42]. However, the surface of WF(PA/B) was denser than WF, which improved the hydrophobicity of the material [43,44].

The cross-section of the poplar can be observed in Figure 5b. There were many pipe holes and other holes in the cross-section of the poplar, and the aperture of some large holes was more than 50 μm. In the cross-section of the biocomposites, the fibres were connected tightly and the holes almost disappeared (Figure 5d,f). Accordingly, it is likely that the cell walls collapsed under pressure and bounded tightly due to the gluing action of lignin. As a result, the biocomposites increased in density and in terms of mechanical properties [45]. It is worth noting that there were either large or small cavities in the cross-section of all biocomposites, except in the case of WF(PA/B). Consequently, it was concluded that the close packing of the fibres caused the increase in tensile strength of WF(PA/B), but also caused its increased brittleness [46,47].

### 3.5. Chemical Properties of Biocomposites

As shown in Figure 7a,b, the phosphoric acid pretreatment was able to remove most of the hemicelluloses, and the wood powder obtained by the ball milling had the lowest hemicellulose content, corroborating the findings shown in Figure 6b. Furthermore, the lightness of all biocomposites was lower than that of poplar [48,49], and that of WF(PA/M) was the lowest. In contrast, the lightness of other biocomposites treated with acid was higher than that of WF (Figure 6a). The results show that lignin melts under high temperature and pressure, forming a carbonised layer on the surface (Figure 7c) [50,51]. However, direct phosphoric acid pretreatment and microwave pretreatment initially dissociate the low-melting-point lignin, producing biocomposites with low content of molten lignin and higher lightness under the same hot-pressing conditions [52,53].

The chemical composition of biocomposites was evaluated by XRD and FT-IR. The Fourier transform infrared (FTIR) spectra of biocomposites and poplar (Figure 6c) show that hot-pressing and acid treatment have an insignificant influence on the destruction of the main components of wood; the characteristic peaks of related functional groups remained. In comparison to poplar, biocomposites has a stronger peak at 1596 cm^−1^ corresponding to the stretching of a carbon-to-carbon double bond conjugated with the benzene ring, indicating that hot pressing could improve the proportion of lignin in materials which was consistent with the result of chemical composition analysis (Figure 6b). The WF(PA/B) shows a weaker peak at 1736 cm^−1^ than that of poplar and other biocomposites, corresponding to the stretching of a carbon-to-oxygen double bond in ketones and esters, which was in concert with the lowest proportion of hemicelluloses being found in WF(PA/M). The decrease of hemicelluloses promoted the closer combination of lignin and cellulose and subsequently improved the mechanical properties of biocomposites (Figure 7c). In short, lignin migrated to the surface of the materials, surface hydrophobicity was enhanced by hot-pressing, hemicellulose content of the wood fibres was significantly diminished by ball milling, and the bonding between the fibres was improved.

The crystallinity of biocomposites had a significant influence on their physical and chemical properties [54]. Two diffraction peaks are associated with cellulose type I crystals (101 and 002) (Figure 6d). The results showed that the cellulose crystal type of biocomposites remained unchanged, proving that acid treatment and hot pressing could not alter the molecular structure of the cellulose, which underlies the mechanical strength of the biocomposites. The relative crystallinity of WF(PA) and WF(PA/M) was lower than that of poplar. This indicated that the dimensional stability of WF(PA) and WF(PA/M) is reduced, resulting in a decrease in the water resistance of the biocomposite [55,56].

### 3.6. Economic Analysis and Future Development

Due to the low price of raw materials and the low-energy preparation process, the cost of the biocomposites prepared in this experiment is greatly reduced. The production cost of the biocomposites (35.21 USD/m^3^) is far lower than that of wood-based panels commonly produced by adhesives in the market, such as particleboard (270.39 USD/m^3^) and medium-density fibreboard (296.78 USD/m^3^) (Figure 8). Furthermore, since no adhesives are used, the biocomposites are environmentally friendly in the production process. At the same time, the biocomposites use waste poplar powder as raw material, achieving 100% product utilisation and recycling rate, in line with the concept of sustainable development. However, the ball milling and hot-pressing take a long time in the production process, so the plate forming time is also lengthened. In addition, hot moulding requires high precision of the mould, so large-scale production needs to be explored. In the future, the production of panels can be expanded, and the production time can be reduced by batch production.

## 4. Conclusions

The poplar powder was pre-treated with phosphoric acid, ball milling, and hot pressing. The physical and chemical properties of adhesive-free biocomposites were evaluated after one-step hot-moulding. WF(PA/B) exhibited the best mechanical properties and thermal stability. The biocomposite fibres were crushed and interconnected tightly through hot pressing, enhancing their mechanical strength. The majority of the hemicelluloses were decomposed with phosphoric acid, which increased the contact between cellulose and lignin and improved the mechanical properties, thermal stability, and hydrophobicity of the biocomposite. Additionally, phosphoric acid concentration may be further investigated in efforts to develop a sustainable and pollution-free biocomposite production method. Overall, the biocomposites produced by this process exhibited high tensile strength and hydrophobicity, revealing a new material for potential use in high-pressure and high-humidity working environments.

## Figures and Tables

**Figure 1 polymers-14-03636-f001:**
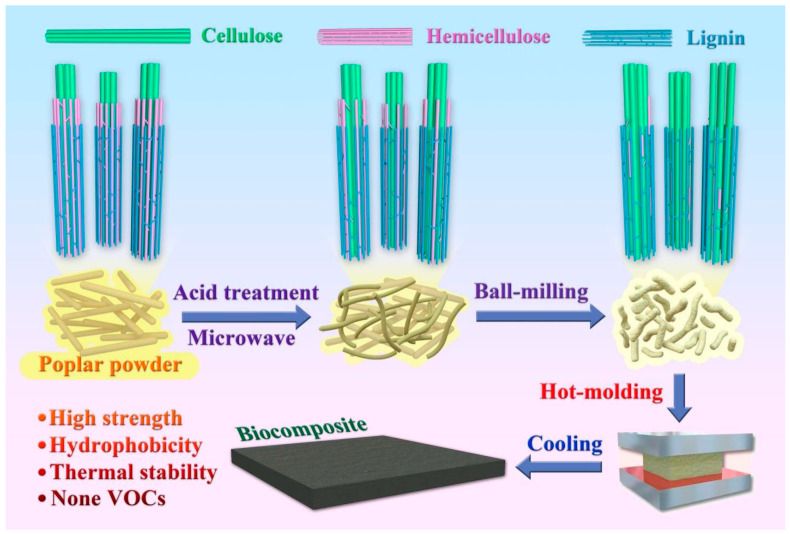
The production process of biocomposites from poplar via pretreatment of phosphoric acid followed by ball-milling and hot-moulding.

**Figure 2 polymers-14-03636-f002:**
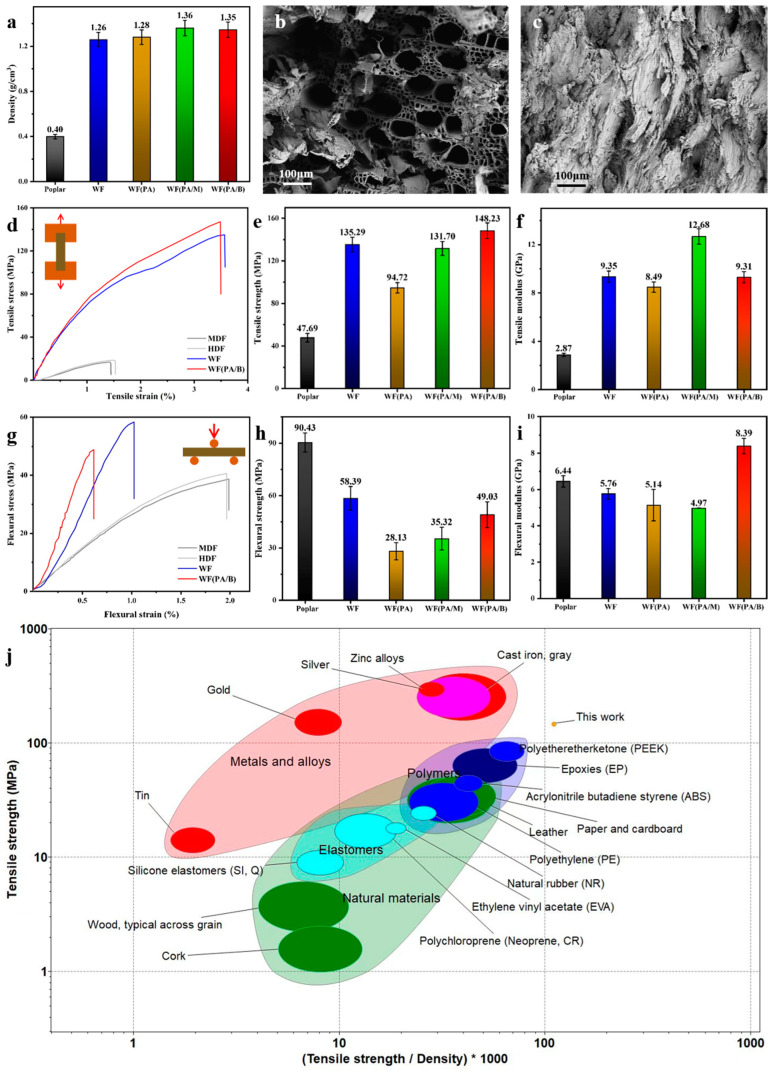
(**a**) Densities of raw material (poplar) and product (biocomposites). (**b**) Cross-section images of the raw material. (**c**) Cross-section images of the best product (WF(PA/B)). (**d**–**f**) Tensile stress–strain curves, tensile strengths, and modulus of samples. (**g**–**i**) Flexural stress–strain curves, flexural strength, and modulus of samples. (**j**) Comparison of several common products and biomaterials with respect to tensile and specific strengths.

**Figure 3 polymers-14-03636-f003:**
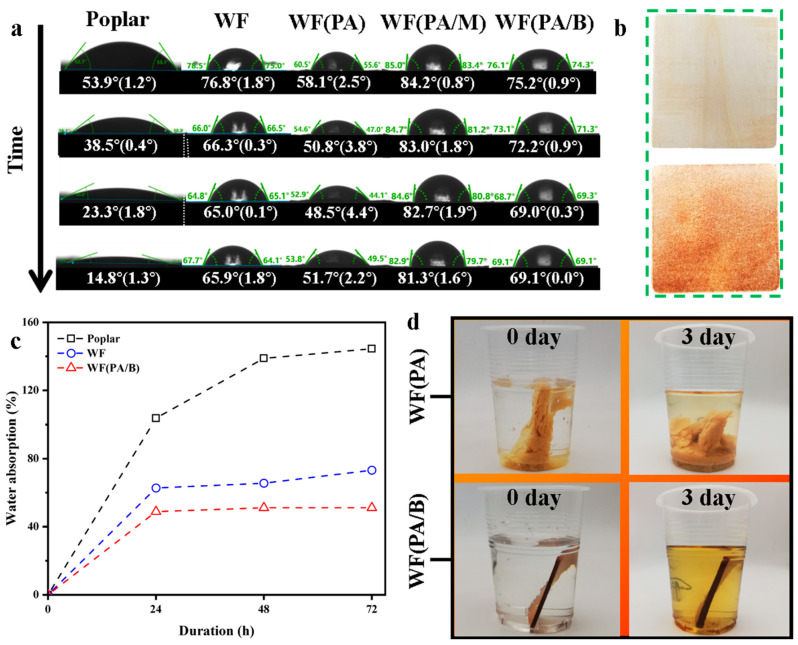
(**a**) Contact angle of poplar and multiple biocomposites at 0 s, 2 s, 5 s and 10 s. (**b**) Image of the surface of poplar and WF(PA/B). (**c**) Water absorption of WF, WF(PA/B) and poplar. (**d**) The water absorption process of WF(PA) and WF(PA/B).

**Figure 4 polymers-14-03636-f004:**
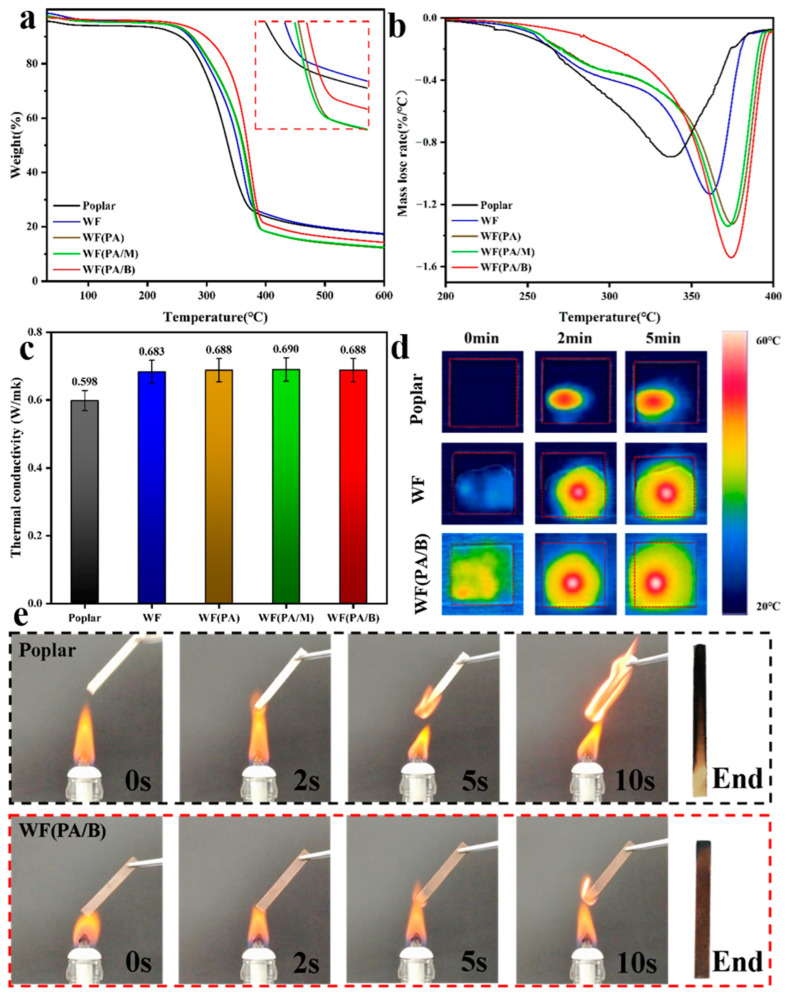
(**a**) TGA analysis of raw material and multiple biocomposite products. (**b**) Results of DTG analysis on raw material and multiple biocomposite products. (**c**) Thermal conductivity of raw material and multiple biocomposites products. (**d**) Thermal imaging of raw material, WF and WF(PA/B). (**e**) Combustion process image of poplar and WF(PA/B).

**Figure 5 polymers-14-03636-f005:**
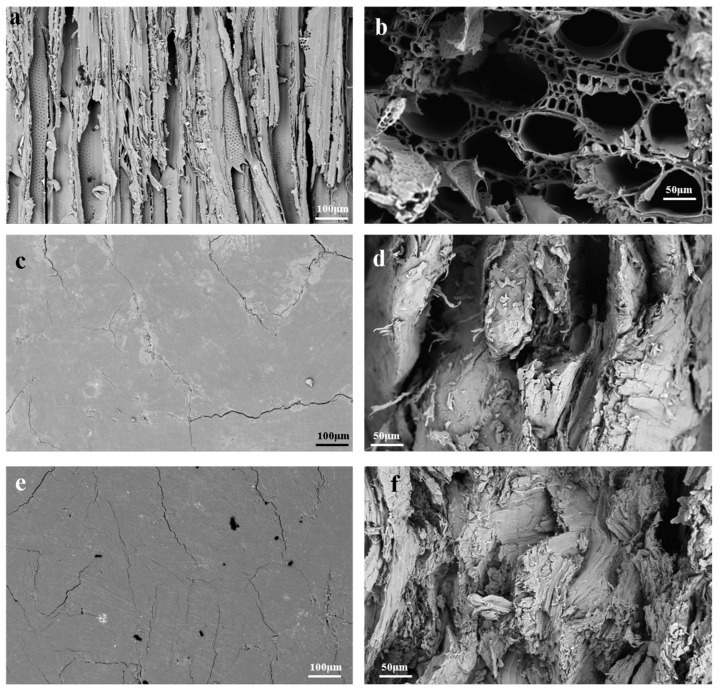
(**a**) Poplar surface viewed at a microscopic scale. (**b**) Cross-sections of poplar microscopic images. (**c**) WF surface viewed at a microscopic scale. (**d**) Cross-sections of WF microscopic images. (**e**) WF(PA/B) surface viewed at a microscopic scale. (**f**) Cross-sections of WF(PA/B) microscopic images.

**Figure 6 polymers-14-03636-f006:**
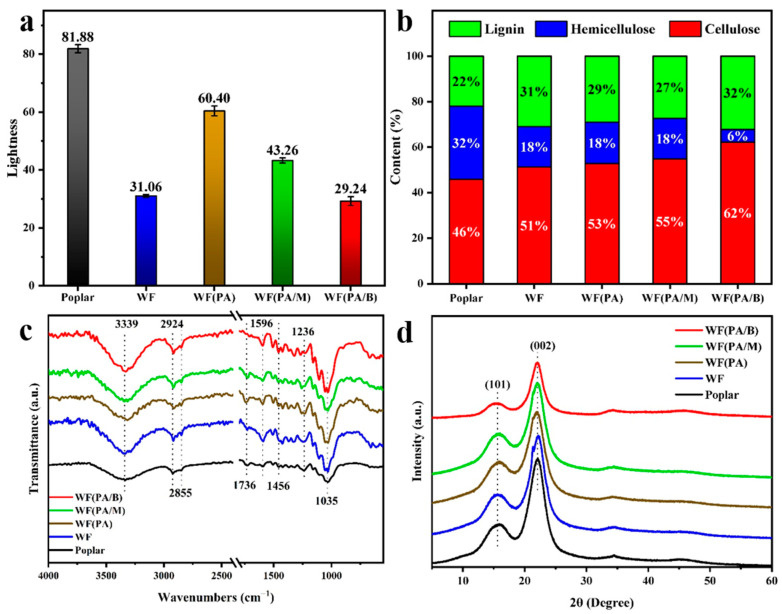
(**a**) Lightness contrast of poplar biocomposites. (**b**) Chemical composition of poplar and biocomposites. (**c**) ATR FT-IR spectra of multiple biocomposites products and raw material. (**d**) XRD patterns of multiple biocomposite products and raw materials.

**Figure 7 polymers-14-03636-f007:**
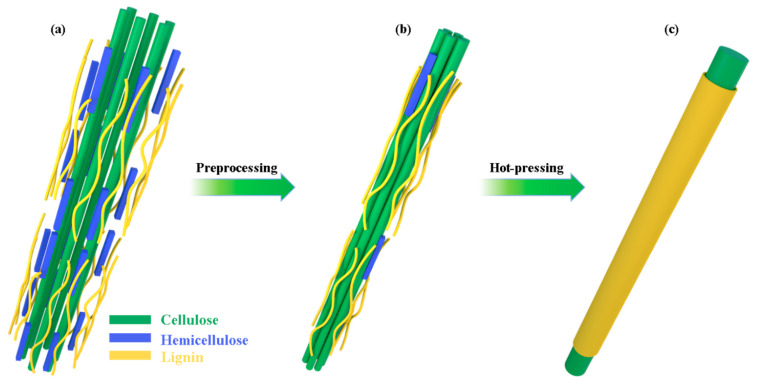
Bonding mechanism of (**a**) Natural bamboo. (**b**) Bamboo powder after preprocessing. (**c**) Biocomposites.

**Figure 8 polymers-14-03636-f008:**
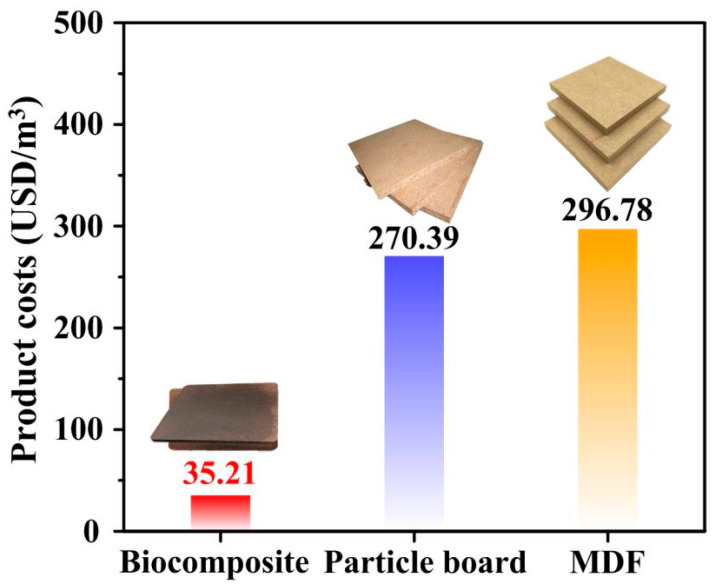
Production costs of biocomposites, particle board and MDF.

## Data Availability

The data presented in this study are available on request.

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
