# Peer review of "New Poplar-Derived Biocomposites via Single-Step Thermoforming Assisted by Phosphoric Acid Pretreatment"

_polymers, 2022, doi:10.3390/polym14173636_

Round 1

Reviewer 1 Report

The manuscript is interesting and contains good data. However, several abbreviations have been added to the manuscript, including WF, MDF, HDF, and more, and the details are missing. These abbreviations are added to Figures, and correlated with the data's physical properties, hydrophobicity of raw material and biocomposites, stability of biocomposites against high temperature, and more sections in the results and discussion. Without, knowing the full form/ meaning of WF, MDF, HDF, and more, is tough to correlate data, understand, and provide scientific comments. Therefore, the editor first should send the manuscript to the authors and ask them for careful editing, particularly for abbreviations. The editor can then send the manuscript for review.

Author Response

Response to the editors and reviewers

Manuscript ID: polymers-1846561

Title: New poplar-derived biocomposites via single-step thermoforming assisted by phosphoric acid pretreatment

Dear Editor,

We thank you very much for the support from the reviewer. We have checked the comments one by one and the responses are shown below. The entire manuscript has been revised according to the editor and reviewers’ comments. The comments of the editor and reviewers are presented below in italics with grey shading, and our responses to those comments are shown in blue font with an underline. The modifications and improvements made to the manuscript are also shown in blue font.

Reviewers' comments:

Reviewer 1: 

The manuscript is interesting and contains good data. However, several abbreviations have been added to the manuscript, including WF, MDF, HDF, and more, and the details are missing. These abbreviations are added to Figures, and correlated with the data’s physical properties, hydrophobicity of raw material and biocomposites, stability of biocomposites against high temperature, and more sections in the results and discussion. Without, knowing the full form/ meaning of WF, MDF, HDF, and more, is tough to correlate data, understand, and provide scientific comments. Therefore, the editor first should send the manuscript to the authors and ask them for careful editing, particularly for abbreviations. The editor can then send the manuscript for review.

Response: We thank the reviewer very much for the comments. We are sorry that our misnomer caused trouble. According to your suggestion, all abbreviations are specified in the article. Meanwhile, the modifications have been made on page 4-5, 7, Line 144-145, 150, 174-178, 181-182, and 209-210, as shown below:

The surface and sectional morphologies of the biocomposites were analysed with the Quanta 200 scanning electron microscope (SEM) manufactured by the FEI company in China.

The biocomposites were examined by an X-ray diffractor manufactured (XRD) by Beijing Purvey General Instruments Co., Ltd. in Beijing, China, to measure the crystallinity.

The highest tensile strength of bio-composite prepared by wood flour with ball milling (WF(PA/B)) was 148 MPa, about three times higher than that of poplar. It is worth noting that the flexural strength of biocomposites prepared by wood flour with phosphoric acid (WF(PA)) was lower than that of biocomposites prepared by wood flour (WF) (Fig. 2h, 2i).

In addition, compared with medium-density fiberboard (MDF) and high-density fiberboard (HDF) (Fig. 2d, 2g), the tensile strength and flexural strength of WF(PA/B) greatly improved.

It is worth noting that the shape of WF(PA) and biocomposite prepared by wood flour with microwave process (WF(PA/M)) changes immediately after immersion in water, as opposed to WF and WF (PA/B), which remained unchanged after immersion for three days (Fig. 3d).

Reviewer 2 Report

Reviewers' comments:

Manuscript Number: polymers-1846561

Title: New poplar-derived biocomposites via single-step thermoforming assisted by phosphoric acid pretreatment

The MS deals with the new poplar-derived biocomposites via single-step thermoforming assisted by phosphoric acid pretreatment. The work is interesting, however, I have several significant concerns to be addressed, my comments and suggestions are listed below:

Comments

-          Section 2.2 & 2.3: Methodology: The author should provide a detailed protocol for all the measurements.

-           I suggest the authors perform the statistical analysis for the data in figure 2, 3, 4 & 6 and mention the significant differences using different letters.

-          Section3. Result and discussion: The author should improve the result and discussion part by explaining the results in detail and discuss the work with the comparison of previous literature reported.

-          There is a lack of discussions and references in section 3, Include more relevant references and discuss, elaborately.

-          English editing is needed for this manuscript. Authors have to carefully check the English grammar throughout the manuscript.

Author Response

Response to the editors and reviewers

Manuscript ID: polymers-1846561

Title: New poplar-derived biocomposites via single-step thermoforming assisted by phosphoric acid pretreatment

Dear Editor,

We thank you very much for the support from the reviewer. We have checked the comments one by one and the responses are shown below. The entire manuscript has been revised according to the editor and reviewers’ comments. The comments of the editor and reviewers are presented below in italics with grey shading, and our responses to those comments are shown in blue font with an underline. The modifications and improvements made to the manuscript are also shown in blue font.

Reviewers' comments:

Reviewer 2: 

The MS deals with the new poplar-derived biocomposites via single-step thermoforming assisted by phosphoric acid pretreatment. The work is interesting, however, I have several significant concerns to be addressed, my comments and suggestions are listed below:

Response: We appreciate the reviewer’s recommendation. Thank you very much!

Section 2.2 & 2.3: Methodology: The author should provide a detailed protocol for all the measurements

Response: We thank the reviewer very much for the comments. We are very sorry that our lack of information has caused you trouble. Meanwhile, the modifications have been made on page 3-4, lines 118-127, and 155-157, as shown below:

2.2. Analysis of essential mechanical and physical characteristics of biocomposite

The water absorption, flexural strength, tensile strength and density of biocomposites were tested by GB/T 17657-2013 standard. The universal testing machine (AGS-X, Shimadzu, Japan) was used to test the tensile and flexural strengths of biocomposites. The tensile and flexural strengths of biocomposites were tested by applying tensile stress and bending force, respectively. The biocomposites were immersed in water, and the changes in weight and thickness were measured every other day until the water absorption was saturated. Each sample was tested three times and the average value was reported. Moreover, the water absorption of the biocomposites was evaluated via complete panels while the biocomposites were sliced 8 mm wide to quantify the mechanical characteristics.

2.3. Chemical structure characterisation

The lignocellulosic of biocomposites was interpreted via Waters E2695 high-performance liquid chromatography. The acid hydrolyzate of biocomposites was analysed by high-performance liquid chromatography (HPLC) to test the content of each component of lignocellulose.

I suggest the authors perform the statistical analysis for the data in figure 2, 3, 4 & 6 and mention the significant differences using different letters.

Response: We thank the reviewer for the comments. We are very sorry that our lack of information has caused you trouble. All of our pictures in this article are for direct comparison and analysis. We have tried the method you recommended before, but we feel that the current form of presentation is relatively better, so please understand our expression. Thank you very much!

Section3. Result and discussion: The author should improve the result and discussion part by explaining the results in detail and discuss the work with the comparison of previous literature reported.

There is a lack of discussions and references in section 3, Include more relevant references and discuss, elaborately.

Response: We thank the reviewer very much for the comments. We are very sorry that our lack of information has caused you trouble. Regarding your two comments, we have made a unified reply. The modifications have been made on page 4-5, 7-8, 10-11, lines 164-167, 176-178, 187-190, 202-205, 241-245, 266-267, 313-320, as shown below:

This proves that the cell wall collapses in the biocomposite, the pores are flattened, the distance between cells decreases, and the tight junctions between fibres form chemical bonds after hot-pressing [21-22].

It is worth noting that the flexural strength of biocomposites prepared by wood flour with phosphoric acid (WF(PA)) was lower than that of biocomposites prepared by wood flour (WF) (Fig. 2h, 2i).

By comparing the specific strength between the prepared biocomposite and other commonly used materials (Fig. 2j), it was observed that the resulted biocomposite had higher specific strength than many other materials such as metal, alloy, bricks, and cement. This indicates that the biocomposites had the conditions for application in structural building materials of specific strength [25-26].

The hot-pressing process of the fibres caused the pores of the biocomposites to become more tightly packed, thereby preventing water from penetrating the pores [24]. At the same time, lignin covered the biocomposite surface during hot-pressing, which hindered water entry to a certain extent [29-30].

There may have been a protective layer formed when the lignin was transferred to the surface of the board during hot pressing. During high-temperature combustion, part of the hemicelluloses was decomposed, and lignin formed a dense carbonisation film on the surface of bio-composites, which hindered the decomposition of fibres from improving the thermal stability and thermal degradation resistance of biocomposites [36-37].

The surface micromorphology of WF(PA) and WF(PA/M) (Fig. S1) show that compared with WF, the fibres treated with phosphoric acid were challenging to combine, and the surface of the board was rougher, which decreased the surface hydrophobicity [39-40]. However, the surface of WF(PA/B) was denser than WF, which improved the hydrophobicity of the material [41-42].

The crystallinity of biocomposites had a significant influence on their physical and chemical properties [40]. Two diffraction peaks are associated with cellulose type I crystals (101 and 002) (Fig. 6d). The results showed that the cellulose crystal type of biocomposites remained unchanged, proving that acid treatment and hot pressing could not alter the molecular structure of the cellulose, which underlies the mechanical strength of the biocomposites. The relative crystallinity of WF(PA) and WF(PA/M) was lower than that of poplar. This indicated that the dimensional stability of WF(PA) and WF(PA/M) is reduced, resulting in a decrease in the water resistance of the biocomposite [52-53].

Additional references:

[21] Wang Q, Xiao S, Shi SQ, Cai L. The effect of delignification on the properties of cellulosic fiber material. Holzforschung 2018;72(6):443-49.

[22] Shi J, Peng J, Huang Q, Cai L, Shi SQ. Fabrication of densified wood via synergy of chemical pretreatment, hot-pressing and post mechanical fixation. Journal of Wood Science 2020;66(1).

[25] Pabón Rojas JJ, Ramón Valencia BA, Bolívar Osorio FJ, Ramirez D. The role of fiber-matrix compatibility in vacuum processed natural fiber/epoxy biocomposites. Cellulose 2021;28(12):7845-57.

[26] Moussa T, Maalouf C, Bliard C, Abbes B, Badouard C, Lachi M, et al. Spent Coffee Grounds as Building Material for Non-Load-Bearing Structures. Materials (Basel) 2022;15(5).

[29] Zhao Y, Xiao S, Yue J, Zheng D, Cai L. Effect of enzymatic hydrolysis lignin on the mechanical strength and hydrophobic properties of molded fiber materials. Holzforschung 2020;74(5):469-75.

[30] Zhao Y, Qu D, Yu T, Xie X, He C, Ge D, et al. Frost-resistant high-performance wood via synergetic building of omni-surface hydrophobicity. Chemical Engineering Journal 2020;385.

[37] Alharbi MAH, Hirai S, Tuan HA, Akioka S, Shoji W. Dataset on mechanical, thermal and structural characterization of plant fiber-based biopolymers prepared by hot-pressing raw coconut coir, and milled powders of cotton, waste bagasse, wood, and bamboo. Data Brief 2020;30:105510.

[39] Arisht SN, Abdul PM, Liu C-M, Lin S-K, Maaroff RM, Wu S-Y, et al. Biotoxicity assessment and lignocellulosic structural changes of phosphoric acid pre-treated young coconut husk hydrolysate for biohydrogen production. International Journal of Hydrogen Energy 2019;44(12):5830-43.

[40] Ma CY, Xu LH, Zhang C, Guo KN, Yuan TQ, Wen JL. A synergistic hydrothermal-deep eutectic solvent (DES) pretreatment for rapid fractionation and targeted valorization of hemicelluloses and cellulose from poplar wood. Bioresour Technol 2021;341:125828.

[41] Duan X, Piao X, Xie M, Cao Y, Yan Y, Wang Z, et al. Environmentally friendly wood-fibre-based moulded products with improved hydrophobicity and dimensional stability. Colloids and Surfaces A: Physicochemical and Engineering Aspects 2021;627.

[42] Wu X, Yang F, Gan J, Kong Z, Wu Y. A Superhydrophobic, Antibacterial, and Durable Surface of Poplar Wood. Nanomaterials (Basel) 2021;11(8).

[52] Lule ZC, Kim J. Surface treatment of lignocellulose biofiller for fabrication of sustainable polylactic acid biocomposite with high crystallinity and improved burning antidripping performance. Materials Today Chemistry 2022;23.

[53] Berthet MA, Commandré JM, Rouau X, Gontard N, Angellier-Coussy H. Torrefaction treatment of lignocellulosic fibres for improving fibre/matrix adhesion in a biocomposite. Materials & Design 2016;92:223-32.

English editing is needed for this manuscript. Authors have to carefully check the English grammar throughout the manuscript

Response: We appreciate the reviewer’s recommendation. Our manuscript has been checked and revised by a native English speaker in terms of the language and style required. The proofreading certificate has been attached to the last page.

Reviewer 3 Report

Extensive editing of English language and style required.

Observations:

Introduction

-Page 2, raw 57: This paper proposes a green biocomposite (self-adhesive materials) as an alternative to traditional biocomposite. What means traditional biocomposite????

-Please replace “hemicellulose” with hemicelluloses, in whole article. Hemicelluloses are a class of polysaccharides in woods and plants that are not cellulose.

Materials and methods.

Section 2.1; 2.2: Please reword, there are too many grammar mistakes.

Results and discussion

-page 4, row 170: It is risky to stipulate that “This indicates that it can be used as a building material to substitute metal in some cases.” More investigations are needed to reach this conclusion.

- Section 3.3, page 7, row 197-207, please rephrase!

- Section 3.5: “The cellulose crystal type of biocomposites remained unchanged, proving that acid treatment and hot pressing could not alter the molecular structure of the wood fiber, which underlies the mechanical strength of the biocomposites”. Please correct: the structure of wood fiber is affected because hemicelluloses are removed, you are probably talking about the cellulose fiber.

Pag 10: row 289-290: “This indicates that the crystallization is destroyed after treatment, which may also be the reason for the poor hydrophobicity of the biocomposite”: Crystallization is not equivalent to crystallinity/structural order.

Author Response

Response to the editors and reviewers

Manuscript ID: polymers-1846561

Title: New poplar-derived biocomposites via single-step thermoforming assisted by phosphoric acid pretreatment

Dear Editor,

We thank you very much for the support from the reviewer. We have checked the comments one by one and the responses are shown below. The entire manuscript has been revised according to the editor and reviewers’ comments. The comments of the editor and reviewers are presented below in italics with grey shading, and our responses to those comments are shown in blue font with an underline. The modifications and improvements made to the manuscript are also shown in blue font.

Reviewers' comments:

Reviewer 3: 

Extensive editing of English language and style required.

Response: We thank the reviewer very much for the comment. Our manuscript has been checked and revised by a native English speaker. The proofreading certificate has been attached to the last page.

Introduction

Page 2, raw 57: This paper proposes a green biocomposite (self-adhesive materials) as an alternative to traditional biocomposite. What means traditional biocomposite????

Response: We thank the reviewer very much for the comment. We apologise for the inconvenience caused by our misinterpretation. The traditional biocomposite is made of wood (its various forms, including fibre, veneer and shaving) as the matrix material, reinforcing materials or functional materials. Traditional biocomposite is commonly referred to as composite material with certain load-bearing or specific properties.

Please replace “hemicellulose” with hemicelluloses, in whole article. Hemicelluloses are a class of polysaccharides in woods and plants that are not cellulose.

Response: We thank the reviewer very much for the comments. We apologise for the inconvenience caused by the improper word. According to your suggestion, we have replaced “hemicellulose” with hemicelluloses throughout this article.

Materials and methods.

Section 2.1; 2.2: Please reword, there are too many grammar mistakes.

Response: We thank the reviewer very much for the comments. We apologise for the inconvenience caused by the improper word. The modifications have been made on page 3-4, Line 102-106, 113, 118-127, as shown below:

2.1. Source of feedstock and its utilisation to produce biocomposites

The poplar powder (40-60 mesh) used in this study was supplied by the Lianyungang timber wholesale market in Northern Jiangsu. Then, the powder was pre-treated under different conditions, such as the phosphoric acid soaking method, microwave treatment after soaking, and ball milling with a phosphoric acid solution (Table S1). First, the poplar wood flour was soaked in 5% phosphoric acid for 3h, and the pre-treated poplar wood powder was rinsed with plenty of water until the residual phosphoric acid was completely removed. Then, powder samples were dried in an oven at about 100 °C for one day. About 8 g of powder were taken from each sample and followed by subjecting to pressurised hot-molding (30 MPa) for 60 min at 186 ℃ with a ZG-50TSD thermocompressor manufactured by Zhenggong Electromechanical Equipment Technology Co., Ltd., located in China, to synthesise a 25 cm2 biocomposites with a thickness of 3.5 mm. Each sample produced three parallel specimens. Furthermore, only one parameter of hot-molding (30 MPa, 186 ℃) was selected as the control variable.

2.2. Analysis of essential mechanical and physical characteristics of biocomposite

The water absorption, flexural strength, tensile strength and density of biocomposites were tested by GB/T 17657-2013 standard. The universal testing machine (AGS-X, Shimadzu, Japan) was used to test the tensile and flexural strengths of biocomposites. The tensile and flexural strengths of biocomposites were tested by applying tensile stress and bending force, respectively. The biocomposites were immersed in water, and the changes in weight and thickness were measured every day until the water absorption was saturated. Each sample was tested three times and the average value was reported. Moreover, the water absorption of the biocomposites was evaluated via complete panels while the biocomposites were sliced 8 mm wide to quantify the mechanical characteristics.

Results and discussion

page 4, row 170: It is risky to stipulate that “This indicates that it can be used as a building material to substitute metal in some cases.” More investigations are needed to reach this conclusion.

Response: We thank the reviewer very much for the comments. We apologise for the inconvenience caused by our misrepresentation. Meanwhile, the modifications have been made on Page 5, lines 187-190, as shown below:

By comparing the specific strength between the prepared biocomposite and other commonly used materials (Fig. 2j), it was observed that the resulted biocomposite had higher specific strength than many other materials such as metal, alloy, bricks, and cement. This indicates that the biocomposites had the conditions for application in structural building materials of specific strength  [25-26].

Section 3.3, page 7, row 197-207, please rephrase!

Response: We thank the reviewer very much for the comments. We apologise for the inconvenience caused by our improper use of words. According to your suggestion, we have replaced “hemicellulose” with hemicelluloses in this paragraph. Meanwhile, the relevant modifications have been made on page 8, lines 221-227, as shown below:

The thermal stability and thermal conductivity of the biocomposites were tested. As shown in Fig. 4a, the poplar weight was first reduced while the WF(PA/B) weight was reduced at higher temperatures in the main thermal decomposition stage (240-380 ℃). This may be due to the lowest hemicelluloses content of the WF(PA/B) (Fig. 6b), in which the hemicelluloses have the lowest degradation stability among cellulose hemicelluloses and lignin [25, 26]. The DTG curve further corroborates this finding, where the poplar containing more hemicelluloses tends to degrade at lower temperatures at 240-380 ℃. The mass-loss rate of poplar was the highest, whereas the WF (PA/B) was the lowest in the temperature range of hemicelluloses (200-280 ℃). The mass-loss rate of WF (PA/B) was the largest at 350-400 ℃, which may be caused by the higher proportion of cellulose and lignin, requiring higher decomposition temperature to degrade these components.

Section 3.5: “The cellulose crystal type of biocomposites remained unchanged, proving that acid treatment and hot pressing could not alter the molecular structure of the wood fiber, which underlies the mechanical strength of the biocomposites”. Please correct: the structure of wood fiber is affected because hemicelluloses are removed, you are probably talking about the cellulose fiber.

Response: We thank the reviewer very much for the comments. We apologise for the inconvenience caused by our unclear expression. Meanwhile, the modifications have been made on page 11, lines 313-315, as shown below:

The crystallinity of biocomposites had a significant influence on their physical and chemical properties [40]. Two diffraction peaks are associated with cellulose type I crystals (101 and 002) (Fig. 6d). The results showed that the cellulose crystal type of biocomposites remained unchanged, proving that acid treatment and hot pressing could not alter the molecular structure of the cellulose, which underlies the mechanical strength of the biocomposites.

Page 10: row 289-290: “This indicates that the crystallisation is destroyed after treatment, which may also be the reason for the poor hydrophobicity of the biocomposite”: Crystallisation is not equivalent to crystallinity/structural order.

Response: We thank the reviewer very much for the comments. We apologise for the inconvenience caused by our misrepresentation. Meanwhile, the modifications have been made on page 11, lines 317-320, as shown below:

The relative crystallinity of WF (PA) and WF (PA/M) was lower than that of poplar. This indicated that the dimensional stability of WF(PA) and WF(PA/M) is reduced, resulting in a decrease in the water resistance of the biocomposite [52-53].

Round 2

Reviewer 3 Report

-